# Prognostication of DNA Damage Response Protein Expression Patterns in Chronic Lymphocytic Leukemia

**DOI:** 10.3390/ijms24065481

**Published:** 2023-03-13

**Authors:** Ti’ara L. Griffen, Fieke W. Hoff, Yihua Qiu, Jan Burger, William Wierda, Steven M. Kornblau

**Affiliations:** 1Department of Microbiology, Biochemistry, and Immunology, Morehouse School of Medicine, Atlanta, GA 30310, USA; 2Department of Internal Medicine, University of Texas Southwestern Medical Center, Dallas, TX 75390, USA; 3Department of Leukemia, The University of Texas MD Anderson Cancer Center, Houston, TX 77030-4009, USA

**Keywords:** CLL, proteomics DDR

## Abstract

Proteomic DNA Damage Repair (DDR) expression patterns in Chronic Lymphocytic Leukemia were characterized by quantifying and clustering 24 total and phosphorylated DDR proteins. Overall, three protein expression patterns (C1-C3) were identified and were associated as an independent predictor of distinct patient overall survival outcomes. Patients within clusters C1 and C2 had poorer survival outcomes and responses to fludarabine, cyclophosphamide, and rituxan chemotherapy compared to patients within cluster C3. However, DDR protein expression patterns were not prognostic in more modern therapies with BCL2 inhibitors or a BTK/PI3K inhibitor. Individually, nine of the DDR proteins were prognostic for predicting overall survival and/or time to first treatment. When looking for other proteins that may be associated with or influenced by DDR expression patterns, our differential expression analysis found that cell cycle and adhesion proteins were lower in clusters compared to normal CD19 controls. In addition, cluster C3 had a lower expression of MAPK proteins compared to the poor prognostic patient clusters thus implying a potential regulatory connection between adhesion, cell cycle, MAPK, and DDR signaling in CLL. Thus, assessing the proteomic expression of DNA damage proteins in CLL provided novel insights for deciphering influences on patient outcomes and expanded our understanding of the potential complexities and effects of DDR cell signaling.

## 1. Introduction

DNA damage recognition and repair protein signaling (i.e., mismatch repair, non-homologous end joining, and homologous recombination) is commonly dysregulated through a multitude of mechanisms (i.e., monoallelic/biallelic loss alterations, methylation, or deletions) or activated in cancer cells to provide a proliferation or survival advantage in the presence of therapeutic agents. A common property of Chronic Lymphocytic Leukemia, an incurable CD19+ CD5+ adult hematological malignancy, is the presence of chromosomal aberrations within the DNA damage and cell cycle signaling pathways [1,2]. Some of the most frequently mutated genes include TP53, ATM, SF3B1, and BRAF. DNA damage signaling is especially altered in a subset of patients harboring TP53, deletion 17P, or ATM mutations, which results in a more aggressive disease and poorer treatment responses [3,4,5]. Patients with TP53 mutations or the 17p deletion are ineligible for receiving the fludarabine, cyclophosphamide, and rituximab (FCR) treatment combination as they quickly become relapsed or refractory; these patients are instead treated with BTK inhibitors (i.e., ibrutinib, acalabrutinib, or zanubrutinib) [6]. Currently, little is known about how these alterations biologically induce therapeutic resistance outside of an MCL1 and BCL-XL-mediated pro-survival response [7]. Additionally, most studies focus on defining the DDR biology of DDR-altered patients, which constitute a small percentage of patients, rather than evaluating DDR protein expression in the entire CLL population.

In this study, we aimed to characterize the proteomic landscape of DNA damage signaling activity in Chronic Lymphocytic Leukemia with Reverse Phase Protein Array. This information will provide biological insights into how DNA damage protein expression is utilized within patients with and without alterations in CLL.

## 2. Results

### 2.1. DDR Protein Expression in Chronic Lymphocytic Leukemia Is Different from Normal and Forms Recurrent Expression Patterns

The level of protein expression of 19 different DDR, and the post-translationally modified (PTM), phosphorylated forms of 5 of these, was measured in 795 Chronic Lymphocytic Leukemia patients and normalized against the expression of normal CD19 cells as shown in Figure 1. Overall, the expression of fifteen proteins lacked variation across the cases and for six proteins (CHEK1.pS296. XRCC, DDB1, RAD51, ATM.pS1981, and CHEK2.pT68) it was predominantly within the normal range (top dendrogram Figure 1), whereas it was universally lower than normal in another six (CHEK2, ERCC1, PDCD1, CHEK1, MSH2, and RAD50), and for three was generally below normal for most cases (MSH6, VCP, and XPA). For the remaining nine proteins only CHEK1.pS345 was above normal levels in nearly all cases, and the other eight (From RAP2 down to SSBP2 in Figure 1) showed heterogeneity with many cases both above and below normal. Thus, for 75% of proteins, abnormal levels of expression were common, highlighting that DDR protein expression in CLL is different from normal B lymphocytes. Using unbiased hierarchical clustering and the progeny clustering algorithm we asked whether recurrent protein expression patterns of DDR proteins occurred in CLL, and if so, what was the optimal number of recurrent signatures of proteins. This identified that recurrent patterns were present, and based on this, the cohort was divided into three clusters, with most patients in clusters C1 (32%) and C3 (61%) and only 7% in C2. Differences in individual proteins’ expression between the three cohorts are shown in Figure 2. Cluster 2 (Pink) was distinguished by significantly higher (*p* ≤ 0.005 after Bonferroni correction for 10 searches) expression of four of five proteins in the top dendrogram, including the PTM form of ATM.pS1981 and CHEK2.pT68 and total DDB1 and Rad51, and markedly lower (*p* < 0.001 after Bonferroni correction for 48 searches) expression of eight of thirteen proteins (all but ATM, CHEK2 ERCC1, ERCC5, and XPF) in the middle dendrogram branch relative to Custer 1 (red) and twelve of thirteen (all but ATM) in cluster 2 (yellow). Cluster 1 differed from cluster 3 (<0.002 after Bonferroni correction for 24 searches) by higher expression of ERCC5 and SSBP2 and lower expression of CHEK1.pS345, CHEK2., ERCC1, and MSH6. Based on the patient cluster protein expression, cluster C2 patients have defective cell cycle checkpoints, base excision repair, and homologous recombination repair signaling compared to clusters C1 and C3. Clusters C1 and C3 have similar DDR pathway activity, except for the expression of SSBP2 observed in C1, which is an additional means of DNA repair.

### 2.2. DDR Cluster Membership Is Independent of Most Traditional Prognostic Features

Next, we compared the expression of other known clinical, laboratory, and molecular predictors of response in CLL between the three clusters. There were no differences between the groups for age, gender, Binet or Rai staging, and no clinically relevant differences for blood counts, B2M or LDH. At the molecular level, there were statistically significant imbalances with prognostically adverse IGHV unmutated, trisomy 12 overrepresented in C1, and deletion 13q underrepresented in C1 and overrepresented in C3. Notably, 17p abnormalities did not differ between the three clusters. However, no molecular event was exclusive to a cluster and all of these molecular events were observed in all clusters (Table 1).

### 2.3. DNA Damage Expression Patterns Are Associated with Adverse Patient Outcomes, Chemotherapy Responses, and Prognostic Factors

To determine the clinical relevance of the DNA damage patient cluster classification, we analyzed associations with known clinical biomarkers and patient time to first treatment and overall survival outcomes. The DNA damage expression patterns were prognostic for survival outcomes but not for time to first or second treatment (Figure 3). Patients within clusters C1 and C2 (median 13.5 years) had similar but significantly shorter (*p* = 0.0001) overall survival compared to C3 (median not reached but >25 years). Therapy for CLL has evolved significantly over the past decade with the advent of Bruton Tyrosine Kinase inhibitors and P13K delta inhibitors, alone or combined with a BCL2 inhibitor (venetoclax), superseding the prior standard of conventional chemotherapy given alone or, more commonly, in combination with anti-CD20 antibodies. We, therefore, queried whether DDR protein expression was prognostic regardless of therapy. There were 121 patients who received the fludarabine–cyclophosphamide–rituximab (FCR) regimen. A total of 108 CLL patients received BTKi therapy (i.e., acalabrutinib, ibrutinib, spebrutinib, or zanubrutinib) as monotherapy (n = 40) or combined with venetoclax (n = 40), or an antibody (i.e., anti-CD20 rituximab), anti-ROR1 cirmtuzumab, or anti-PDL1 nivolumab (n = 10), or combined with chemotherapy (fludarabine, cyclophosphamide, or obinutuzumab (iFCG therapy), n = 18). Similar to the results from the entire CLL population, DNA damage protein expression patterns were prognostic (*p* = 0.03) for overall survival in patients treated with FCR but not in patients treated with BTK inhibitors (Figure 4); although, five of six deaths in BTKi treated patients were in C1 and C2. FCR-treated cluster C1 patients had significantly shorter survival (median 16 years) than C3 patients (median 22.6 years). FCR-treated patients within cluster C3 were associated with several good prognostic factors including lower proportions of death (8.4%), trisomy 12 (7%), higher proportions of deletion 13Q (47%), as well as having a lower expression of CD79b (*p* = 0.02), CD38 (*p* < 0.001), and CD22 (*p* < 0.001). There were no associations observed with age, gender, race, or stage. In the univariate analysis of survival time, the DNA damage clusters and several known cytogenetic aberrations (17p, T12, 13q, and 11q), biomarkers (ZAP70 and IGHV status), staging, and gender were predictive of survival (Table 2). When these variables were assessed together in a multivariate analysis, the DNA damage clusters and ZAP70 status were independent predictors of overall survival, but stage, neither Binet or Rai staging, nor any of the recurrent chromosomal events or IGHV mutation status remained in the final model.

### 2.4. Prognostication of Individual DNA Damage Member Proteins

Next, we assessed associations between individual DNA damage proteins and patient outcomes within the overall population and FCR-treated patients with cox hazard analyses. Several proteins (CHEK1, CHEK1.pS296, CHEK2.pT68, DDB1, PDCD1, RAD51, RPA32, SSBP2, and VCP) were predictive of overall survival (Figure 5 and Appendix A). Notably, total CHEK1, CHEK2.pT68, DDB1, PDCD1, RAD51, and SSBP2 expression negatively affected survival time, whereas CHEK1.pS296, RPA32, and VCP expression had a positive prognostic relationship. Regarding time to first treatment, ATM.pS1981, DDB1, ERRC1, and RAD50 expressions were negative predictors, whereas CHEK1.pS296 was positive. When this analysis was repeated in only FCR-treated patients, none of the individual DNA damage proteins were associated with survival or time to second treatment.

### 2.5. Differential Expression of DNA Damage Protein Expression Groups Reveals Altered Utilization of Adhesion, Cell Cycle, and MAPK Signaling

To investigate the biology that may drive differences in the DNA damage cluster survival outcomes, differential expression analysis was performed between the clusters and the normal CD19 controls for all 384 proteins in our RPPA dataset. Many proteins were significantly differentially expressed compared to the controls (129 for C1, 146 for C2, and 129 for C3). Among these lists, 90 proteins were commonly differentially expressed between the clusters (Appendix A). Notably, an enrichment analysis revealed that the common altered proteins were over-represented with adhesion and cell cycle proteins. The adhesion and cell cycle proteins and their average fold changes across the clusters are denoted in Figure 4A. Most of the cell cycle and adhesion proteins have lower expression in CLL patients, whereas S1004A, WEE1.pS642, and PXN were overexpressed. When evaluating differentially expressed proteins among the clusters, 257, 132, and 255 proteins were altered when comparing C2 to C1, C3 to C1, and C3 to C2, respectively. Enrichment analysis of the altered proteins revealed that there were not any protein functional groups over-represented within the first two comparisons, but MAPK signaling proteins were over-represented in the C3 and C2 comparison. Several MAPK proteins had lower expression in C3, with the exception of five proteins (MAPK1, MAPK1.3.pT202.Y204, MAPK8.pT183.p185, and MAPK9) (Figure 6).

## 3. Discussion

In this study, we quantified 24 total and phosphorylated DDR proteins in CLL patient samples using RPPA and observed that 75% of these proteins had expression levels that were significantly different from normal, with half of the abnormally expressed proteins demonstrating very low expression and the other half showing a heterogenous expression that was both above and below normal. This demonstrates that DDR protein expression is abnormal in CLL cells. Several DNA damage response genes (i.e., ATM, TP53, CHEK1, CHEK2, ERCC4, BRCA1, FANCA, MSH4, SMC1A, RAD50, and MCM3) have previously been shown to be mutated frequently in CLL and to be associated with advanced tumor evolution, immune surveillance escape, and adverse patient outcomes and chemotherapy responses [8,9,10,11,12,13]. Currently, most CLL DNA damage studies are focused on the mutational landscape from whole exome sequencing or gene expression studies that focus on exploring the biology of TP53 or ATM alterations. Exploring the DDR expression patterns at the proteomic level could bring new insights into the final biological consequences of these pathways being altered or unaltered and could be used to propose novel targets for these groups that are likely to relapse or become refractory to the standard of care.

We found that three DNA damage response protein expression patterns exist in CLL and that these are predictive of overall survival and response to the FCR regimen. Notably, cluster C1 patients had the poorest survival with an elevated expression of ERCC5 and SSBP2 and a lower expression of CHEK1.pS345, CHEK2.pS345, ERCC1, and MSH6 compared to the good prognostic cluster C3. These patterns are also not exclusive to known CLL DDR chromosomal aberrations such as the TP53, 11q, and 17p deletions. Our findings are congruent with a previous study that characterized and clustered CLL patients into a genomic unstable subtype that had uniform gene expression patterns within TP53 altered and wildtype patients [14]. These findings further implicate that there are other factors that are determinants of CLL DNA damage response activity (i.e., chemotherapy toxicity-induced genome instability) and that the consequences of the DDR alterations may result in advantageous biological changes beyond the DDR pathways.

Our initial observation of the protein expression in the poor prognostic group led us to hypothesize that CHEK, excision repair, (ERCC1 and ERCC5), and SSBP2 proteins could be predictive of CLL patient outcomes. We found that total CHEK1, CHEK1.pS296, CHEK2.pT68, ERCC1, and SSBP2 were predictive of survival and time to first treatment, with CHEK1.pS296 being a positive prognostic factor. Several studies have proposed and shown the efficacy of inhibiting CHEK1 in TP53 wildtype and deleted in vitro and in vivo CLL cells, which could be considered as an additional combination treatment option [15,16,17]. Our patient C1 cluster data suggests the importance of excision repair proteins in long-term CLL survival and chemotherapy responses. A previous study found that an aspect of fludarabine’s mechanism of action is the suppression of ERCC1-mediated DNA damage repair, which could explain the decreased FCR efficacy and survival in cluster C1 patients [18]. ERCC1 is an excision repair protein that dimerizes with XPF to mediate nucleotide excision repair and double-strand break repairs [19]. Higher expression of this protein has been associated with platinum agent resistance; however, our results show the opposite effect for purine analog-based agent fludarabine [19,20,21]. FCR therapy efficacy is dependent upon CLL cells having an efficient DNA damage response to trigger apoptosis. Since Cluster C1 has absent or low expression of several cell cycle checkpoint proteins (ATM, RAD50, CHEK1, CHEK2, RPA2, and TP53BP1), FCR treatment cannot induce its mechanism of action. 

To identify additional proteomic signaling consequences of the DDR protein expression patterns, we performed differential expression and enrichment analyses of all proteins in our CLL RPPA dataset. When compared to normal controls, all DDR patient clusters consistently had lower adhesion and cell cycle protein expression. Previously, a study confirmed that CLL epithelial-mesenchymal transition (EMT) and DDR signaling are part of a negative feedback loop. EMT transcription factors can downregulate p53 activity to suppress DDR and the reverse can occur [22]. Perhaps our observations confirm that this feedback loop may exist within CLL cell biology as most of the CLL patients had normal/above normal levels of DDR protein activity. Additionally, we observed differential expression between the clusters with the prognostically favorable cluster C3 patients having a lower expression of MAPK proteins and potentially less activation of DDR by MAPK, compared to the more aggressive groups. MAPK signaling is one of the key drivers of CLL growth and proliferation [23]. Constitutive MAPK signaling results in the activation of the DNA damage response because of replication stress and is activated to promote DDR in cisplatin-resistant melanoma cells [24]. Since patient cluster 3 has lower MAPK signaling, their CLL cells have less proliferative potential, have lower DDR responsiveness activity when treated with FCR long-term, and have an overall better prognosis.

Limitations of this study include having a limited number of modified proteins within our dataset and the lack of in vitro or in vivo studies to validate the cell signaling consequences of the DDR protein expression groups. Future studies should be designed to address any biological hypotheses derived from our initial findings including looking into DDR pathway utilization in CLL single-cell populations. However, the information from this study can be used to broaden the spectrum of patients that may benefit from FCR treatment outside of TP53, ATM, and 17p alterations. Overall, our study provided novel information on the spectrum of DDR pathway activity that occurs in CLL from a total and phosphor-protein perspective, the clinical relevance of the expression subgroups and individual proteins, and provided implications for cell signaling that may be interconnected with DDR protein activity; upon further investigation, this information could lead to the discovery of additional therapeutic targets for optimizing the current treatment paradigm for relapsed and refractory patients.

## 4. Materials and Methods

Frozen (n = 727) and fresh (n = 68) blood (n = 743) and bone marrow (n = 52) samples were acquired from patients diagnosed with CLL (n = 795) at the MD Anderson Cancer Center (MDACC) between 2005 and 2019. The samples were collected under protocols Lab01-473, LAB03-0893, Lab 04-0678, Lab08-0431, and Lab07-0719 in accordance with protocols approved by the Institutional Review Board (IRB) of M.D. Anderson Cancer Center. Informed consent was obtained in accordance with the Declaration of Helsinki. Complete details on the clinical, laboratory, and molecular features of these patients, and the treatments used can be found in our prior manuscript [25]. Samples were collected within a year (n = 360), five years (n = 286), ten years (n = 90), after 10 years (n = 52), and more than 20 years (n = 7) after initial diagnosis. These include patients who were never treated (n = 476) or treated within 100 days (n = 26), a year (n = 41), 1–2 years (n = 42), 2–5 years (n = 115), and more than five years (n = 92) after diagnosis. In our prior CLL proteomics manuscript, we confirmed that sample collection intervals (diagnosis to sample), organ (PB or BM), processing of fresh vs. cryopreserved cells, and treatment status prior to collection did not bias the data. 

### 4.1. RPPA Methodology

RPPA was used to create proteomic profiles of 795 patient samples and 5 normal peripheral blood lymphocyte CD19+ controls. Frozen samples were initially processed in the same manner except that they were later thawed, placed in fresh media, layered on Ficoll, centrifuged to remove dead cells, washed with PBS, and centrifuged and counted. The cells were lysed to produce whole cell lysates as previously described and normalized to a concentration of 1 × 104 cells/μL [20]. Five serial two-fold dilutions (1:1, 1:2, 1:4, 1:8, 1:16) of each patient, cell line, or control sample were printed onto slides. The slides were probed with 384 strictly validated primary antibodies and a secondary antibody conjugated to an infrared molecule to amplify and detect the signal. Stained slides were quantitated using Microvigene software (Version 3.4, Vigene Tech, Carlisle, MA, USA).

### 4.2. Data Processing, Normalization, and Quality Control

The SuperCurve R package was utilized to calculate a single value of protein concentration from the five serial dilutions on a log 2 scale [21]. The quality of the staining procedure was further assessed by examining the SuperCurve images and identifying and eliminating slides without sufficient variation in signal or which lacked the expected sigmoidal curve. Loading control and topographical normalization procedures were performed to account for protein concentration and background staining variations. The data were normalized by subtracting the median of the rows and columns across all samples to ensure that sample protein expression estimated from different slides could be compared [22]. Lastly, the median of CD19 control proteins was subtracted to normalize values to a normal median of zero enabling recognition of whether expression in the patient was within, above, or below that of the normal.

### 4.3. Statistics

DNA damage patient expression group numeric and categorical clinical associations were assessed with a Kruskal–Wallis test and chi-square statistical tests using a *p*-value of < 0.05. Clinical outcome associations were tested using a Kaplan–Meier log-rank and cox hazard tests using a *p*-value of < 0.05. Differential expression analysis was performed with an ANOVA test followed by FDR *p*-value correction. A functional enrichment test with FDR-corrected *p*-values was performed on the differentially expressed proteins.

## Figures and Tables

**Figure 1 ijms-24-05481-f001:**
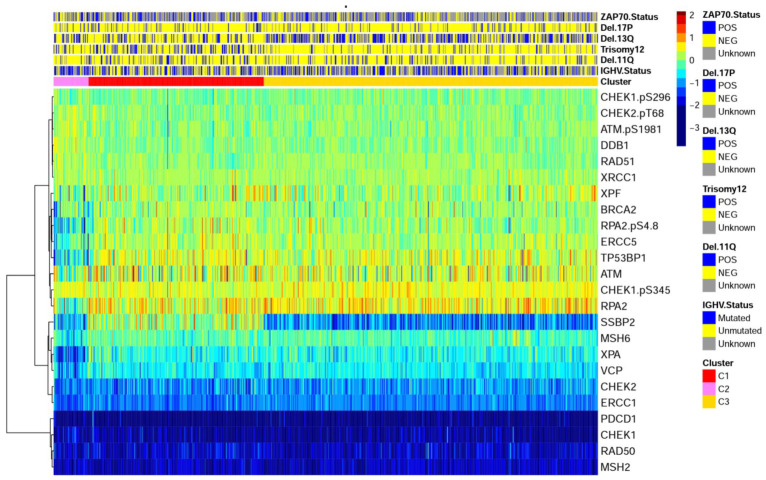
Heatmap of DNA damage protein expression patterns in CLL. CLL patients were clustered based on the range of expression (Log 2 scale, ranging from −4 to +2) of 24 DNA damage proteins in the RPPA dataset. Patient clusters are shown in the annotation immediately above the heatmap (C1-red, C2-pink, and C3-yellow). Selected recurrent chromosomal abnormalities and IGHV mutation status are shown as annotations above the cluster membership, with blue indicating the presence of that event, and yellow indicating its absence.

**Figure 2 ijms-24-05481-f002:**
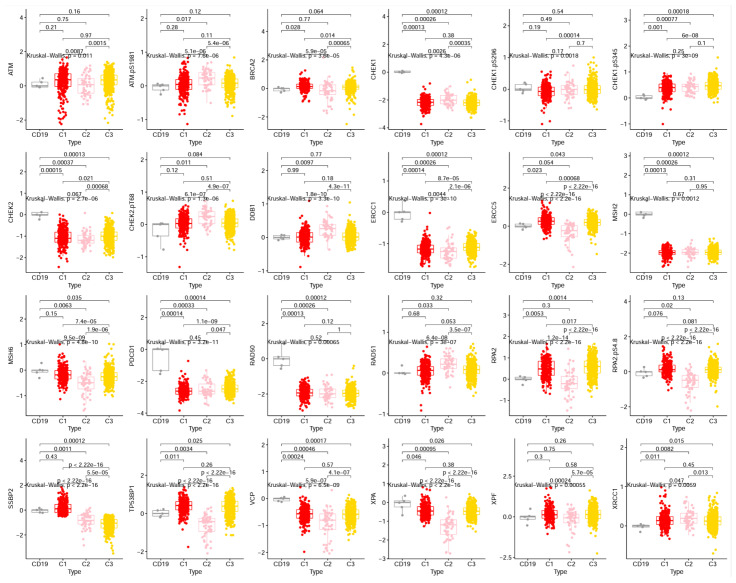
Expression of DDR proteins within and between clusters, relative to normal B-lymphocytes. The boxplot for each of the 24 DDR proteins analyzed in normal CD19+ lymphocytes (Gray) and the three DDR clusters (1 is red, 2 is pink, and 3 is yellow) is shown, along with *p*-values for each possible comparison.

**Figure 3 ijms-24-05481-f003:**
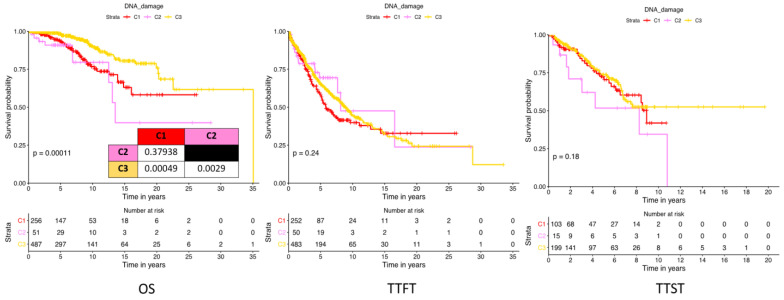
Effect of DDR protein cluster membership on the outcome. Kaplan–Meier plots of the association between the three DDR protein clusters on overall survival (**Left**), time to first treatment (**Center**), and time to second treatment (**Right**) in all CLL patients is shown. For overall survival, comparisons between patients in C1 and C2 were statistically similar (BH *p*-value 0.38) but were both distinct from C3 (vs. C1, BH *p*-value < 0.001 and vs. C2, *p*-value 0.002). For the time to first and second treatments, there were no significant differences.

**Figure 4 ijms-24-05481-f004:**
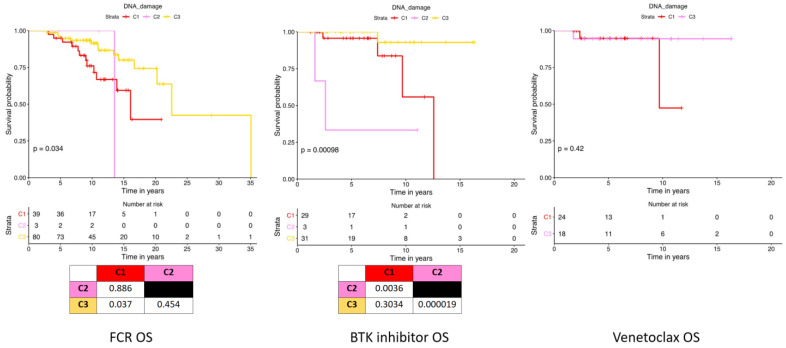
Effect of DDR protein cluster membership on the outcome for different treatments. Kaplan–Meier plots of the association between the three DDR protein clusters on overall survival for patients treated with the fludarabine–cyclophosphamide–rituximab (FCR) regimen (**Left**), with a Bruton Tyrosine Kinase (BTK) inhibitor (**Center**), or with the BCL2 inhibitor venetoclax (**Right**) are shown. For those treated with FCR, C1 patients had shorter OS compared to C3 patients. Notably, the number of FCR-treated C2 patients was too small to draw conclusions. For BTKi, C2 was significantly inferior to C1 (*p* = 0.0036) or C3 (*p* = 0.000019) but C1 and C2 were statistically similar. DDR cluster membership was not prognostic for those treated with venetoclax.

**Figure 5 ijms-24-05481-f005:**
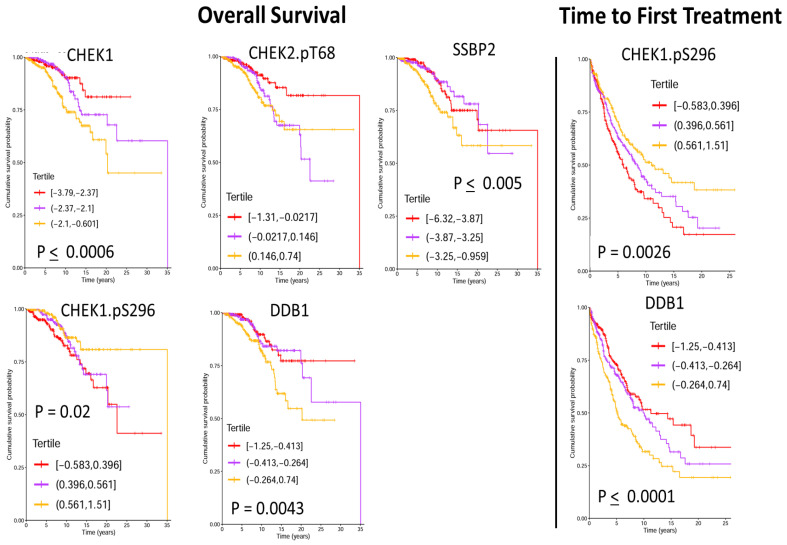
DDR proteins that are individually prognostic of overall survival or time to first treatment. The Kaplan–Meier plots of five proteins that were individually prognostic of overall survival and the two proteins individually prognostic of the time to first treatment in CLL are shown. All proteins were divided into terciles with the lowest expression in red, middle in purple, and highest in yellow.

**Figure 6 ijms-24-05481-f006:**
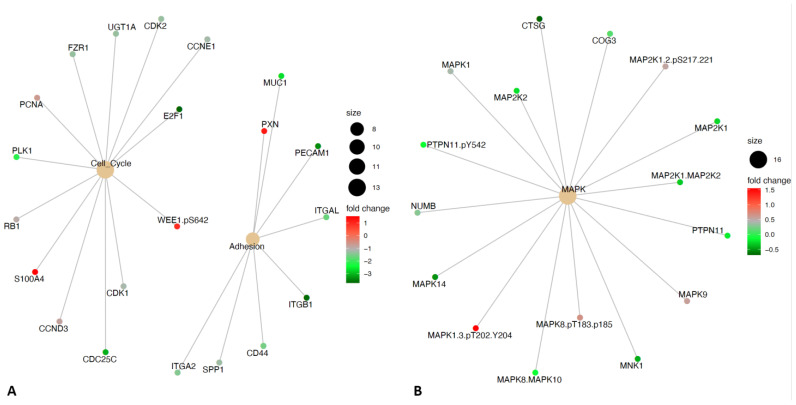
Network plots of Enriched Differentially Expressed Protein Functions when comparing all patient DDR clusters to CD19 controls (**A**) and when comparing the good prognostic cluster C3 to the poor clusters (**B**). Large nodes denote the functional groups and small nodes denote the proteins differentially expressed within each function. In figure (**A**), the color of the protein nodes depicts the average log fold change in expression across all clusters compared to the controls. In figure (**B**), the color scale represents the fold change of cluster C3 patients’ differentially MAPK signaling proteins when compared to C2.

**Table 1 ijms-24-05481-t001:** DNA Damage Group Demographics and Clinical Information. A list of clinical characteristics is displayed in the first column, followed by numeric/categorical calculations for all CLL patients (column 2) and DNA damage patient groups (columns 3–5). In columns 2–5, the mean and standard deviations are displayed for numeric traits and percentages are displayed for categorical traits. Calculations were made based on information available in the dataset for each trait. *p*-values are displayed in the last column from chi-square (categorical) and Kruskal–Wallis tests (numeric) for the groups. Significant associations are in bold.

	TOTAL		C1	C2	C3	*p*-Value
Number	795		256	51	488	
**Age (Mean +/- STD)**		65 (±9.8)	65 (±11)	65 (±11)	65 ± 11	0.91
**Vital Status (Dead)**	88	11.1%	14.8%	17.6%	8.4%	**0.02**
**Race**	**Number**	**Percentage**				0.97
Asian	7	0.9%	0.4%	2.1%	1.1%	
Black	33	4.2%	5.2%	2.1%	4.0%	
Hispanic	22	2.9%	2.8%	2.1%	2.9%	
White	710	92.0%	91.5%	93.6%	92.0%	
**Gender**						0.12
Female	310	39.0%	35.2%	52.9%	39.5%	
Male	485	61.0%	64.8%	47.1%	60.5%	
**Binet Stage**						0.54
A	478	478 (61.0%)	59.5%	29 (58.0%)	62.0%	
B	71	71 (9.06%)	7.9%	2 (4.00%)	10.2%	
C	235	235 (30.0%)	32.5%	19 (38.0%)	27.8%	
**Rai Stage**						0.61
0	268	34.2%	34.5%	38.0%	33.6%	
I	234	29.8%	27.8%	22.0%	31.7%	
II	47	6.0%	5.1%	2.0%	6.9%	
III	132	16.8%	17.9%	28.0%	15.1%	
IV	103	13.1%	14.7%	10.0%	12.7%	
**Biomarkers**						
**IGHV Status (Unmutated)**	**280**	48.6%	58.3%	25.0%	45.7%	**0.002**
**ZAP70**	**189**	50.3%	59.3%	40.5%	47.1%	0.10
**SF3B1**	**34**	16.1%	19.8%	14.3%	13.8%	0.73
**Cytogenetic abberations**						
**Deletion 11Q**	**100**	14.1%	19.0%	6.5%	12.2%	0.05
**Deletion 13Q**	**273**	38.4%	22.0%	37.0%	47.3%	**<0.001**
**Trisomy 12**	**109**	15.3%	30.0%	17.4%	7.2%	**<0.001**
**Deletion 17P**	**68**	95.6%	10.8%	13.0%	8.6%	0.67
**TP53**	**34**	4.3%	4.3%	2.0%	4.5%	0.866
**No Abberations**	**165**	23.2%	20.7%	28.3%	24.0%	0.65
**Lab Tests**	**Units**					
**PB Platelets**	**K/uL**	190 (±72)	190 (±75)	220 (±77)	190 (±70)	**0.03**
**Hemoglobin**	**g/dL**	13 (±1.8)	13 (±2.0)	14 (±1.5)	14 (±1.7)	0.90
**Serum B2M**	**mg/L**	2.8 (±1.8)	2.7 (±1.4)	2.2 (±1.0)	2.8 (±2.0)	0.07
**Serum LDH**	**IU**	480 (±240)	490 (±300)	520 (±210)	460 (±200)	0.27
**Lymphocytes**	**K/uL**	38 (±54)	42 (±61)	18 (±19)	38 (±51)	**0.02**
**Immunophenotypic Markers**						
**CD5**	**% cells positive+**	94 (±11)	93 (±9.5)	94 (±4.0)	94 (±12)	0.15
**CD19**		81 (±15)	82 (±16)	76 (±16)	82 (±14)	0.09
**CD20**		78 (±20)	78 (±21)	79 (±22)	78 (±19)	0.69
**CD22**		63 (±39)	68 (±38)	74 (±37)	59 (±40)	**<0.001**
**CD23**		87 (±18)	86 (±19)	85 (±20)	88 (±17)	0.99
**CD38**		24 (±27)	33 (±31)	23 (±29)	19 (±23)	**<0.001**
**CD79b**		43 (±38)	48 (±33)	40 (±36)	40 (±40)	**0.02**

**Table 2 ijms-24-05481-t002:** Univariate and Multivariate Analysis. Variables that were statistically prognostic of survival in univariate analysis, at a *p*-value of < 0.01 to account for multiple comparisons, are shown on the left side. This includes membership in DDR clusters 1, 2, or 3. These variables were then simultaneously compared in multivariate analysis with only the DDR protein cluster membership and ZAP70 status remaining as significant independent predictors of the outcome.

	Univariate Overall Survival				Multivariate Overall Survival			
Variable	Est.	2.50%	97.50%	*p*-Value	Est.	2.50%	97.50%	*p*-Value
DDR Cluster 1	6.53	5.82	7.24	*p* < 0.01	8.32	4.01	12.62	***p* < 0.01**
DDR Cluster 2	6.37	4.77	7.96	*p* < 0.01	7.04	2.39	11.7	***p* < 0.01**
DDR Cluster 3	7.73	7.22	8.25	*p* < 0.01	9.3	5.28	13.32	***p* < 0.01**
Gender Male	6.98	6.46	7.5	*p* < 0.01	−0.69	−2.07	0.69	0.33
Binet Stage B	9.62	8.28	10.96	*p* < 0.01	2.17	−0.72	5.05	0.14
Binet Stage C	5.97	5.23	6.71	*p* < 0.01	−0.52	−3.59	2.55	0.74
Rai Stage I	8.91	8.17	9.64	*p* < 0.01	1.7	−0.2	3.59	0.08
Rai Stage II	8.49	6.85	10.13	*p* < 0.01	1.46	−1.98	4.89	0.4
Rai Stage III	5.58	4.6	6.56	*p* < 0.01	−1.01	−2.83	0.81	0.27
Rai Stage IV	6.47	5.37	7.58	*p* < 0.01	0.01	−1.83	1.86	0.99
Del_11QPOS	6.47	5.38	7.57	*p* < 0.01	−1.43	−5.2	2.34	0.46
Del_13QPOS	6.79	6.13	7.46	*p* < 0.01	−1.47	−5.18	2.23	0.44
Del_17PPOS	7.41	6.08	8.74	*p* < 0.01	−0.52	−4.44	3.41	0.8
T12POS	5.67	4.63	6.71	*p* < 0.01	−2	−5.78	1.79	0.3
No Major Mutation	6.58	6.12	7.05	*p* < 0.01	−0.9	−4.79	3	0.65
IGHV Unmutated	6.32	5.66	6.99	*p* < 0.01	0.31	−1.18	1.79	0.68
Zap70POS	6.28	5.53	7.03	*p* < 0.01	−1.56	−2.99	−0.13	**0.03**
Binet Stage A and Rai Stage 0 used as comparator								

## Data Availability

https://www.leukemiaatlas.org (accessed on 6 February 2023).

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
