# Peer review of "Prognostication of DNA Damage Response Protein Expression Patterns in Chronic Lymphocytic Leukemia"

_ijms, 2023, doi:10.3390/ijms24065481_

Round 1
Reviewer 1 Report
The paper 'Prognostication of DNA Damage Response Protein Expression Patterns in Chronic Lymphocytic Leukemia' by Griffen et al. describes the identification of three protein expression patterns by quantifying 24 total and phospho DNA Damage Repair proteins in Chronic Lymphocytic Leukemia. Patients within two clusters had poorer survival outcomes and responses to fludarabine, cyclophosphamide and rituximab in comparison to patients included in cluster C3. Nine of the DDR proteins were prognostic for predicting overall survival and/or time to first treatment. The levels of expression of cell cycle and adhesion proteins were lower in clusters in comparison to normal CD19 controls, and patients included in cluster C3 had lower expression of MAPK proteins compared to the patients within clusters C1 and C2. The authors conclude that the proteomic expression of DNA damage proteins in chronic lymphocytic leukemia may help to determine the influences on patient outcomes.
The paper is interesting and deserves publication in International Journal of Molecular Sciences. However, I only propose minor changes.
Check the abstract (lines 22-25) ‘When looking for…… were cell cycle and adhesion proteins’. Also, consider to avoid abbreviations in the abstract.
Keywords are missing.
Revise the abbreviations PDCD1 (line 64), RAP2 (line 68).
Revise the proteins with high expression in cluster 2 (line 79:CHEK2.p345?) and the difference between cluster 1 and cluster 3 (line 85) and also check the label in Figure 2.
Correct the clinical information included (lines 26 and 32) and the volume units in Table 1.
Check the Table 1 and the supplemental Table 1. Change commas by points in supplemental tables.
Discussion section: check line 241.
Revise the number of cells (line 322).
Author Response
Reviewer 1
The paper 'Prognostication of DNA Damage Response Protein Expression Patterns in Chronic Lymphocytic Leukemia' by Griffen et al. describes the identification of three protein expression patterns by quantifying 24 total and phospho DNA Damage Repair proteins in Chronic Lymphocytic Leukemia. Patients within two clusters had poorer survival outcomes and responses to fludarabine, cyclophosphamide and rituximab in comparison to patients included in cluster C3. Nine of the DDR proteins were prognostic for predicting overall survival and/or time to first treatment. The levels of expression of cell cycle and adhesion proteins were lower in clusters in comparison to normal CD19 controls, and patients included in cluster C3 had lower expression of MAPK proteins compared to the patients within clusters C1 and C2. The authors conclude that the proteomic expression of DNA damage proteins in chronic lymphocytic leukemia may help to determine the influences on patient outcomes.
The paper is interesting and deserves publication in International Journal of Molecular Sciences. However, I only propose minor changes.
RESPONSE: We have added line numbering starting from the title page forward for ease of review. However, our line numbers do not align with those added by the journal
Check the abstract (lines 22-25) ‘When looking for…… were cell cycle and adhesion proteins’.
RESPONSE: Duplicated words removed.
Also, consider to avoid abbreviations in the abstract.
RESPONSE: If the editors will expand the word limit we can spell out DDR (DNA damage repair) in the 4 places where we used this acronym I have changed FCR to Fludarabine,Cyclophosphamide,Rituximab
Keywords are missing.
RESPONSE: Added CLL, DDR, Proteomics
Revise the abbreviations PDCD1 (line 64 Line 88 in our revision), RAP2 (line 68 Line 90 in our revision).
RESPONSE: All the protein/gene names are supposed to be HUPO but we note that RPA2 is listed by its alternate name RAP32. Changing that would require changing every figure, which would entail a significant amount of work. We have noted on line 57 of the revision “From RAP2 (shown by its alternate name RPA32…” to clarify. We have fixed the typo and changed PCDC1 to PDCD1.
Revise the proteins with high expression in cluster 2 (line 79: CHEK2.p345?) and the difference between cluster 1 and cluster 3 (line 85)
RESPONSE: In our revision lines 101 (CHEK2.pS345 and CHEK2.pT68) and 106 (CHEK2.pS345and CHEK2) have been corrected.
and also check the label in Figure 2.
RESPONSE: We have looked at the label for figure 2 and do not see anything incorrect. If the reviewer can further detail their concern we will try to address it .
Correct the clinical information included (lines 26 (SF3B1?)and 32 (TP53?)) and the volume units in Table 1.
RESPONSE: We do not know if the reviewer started the line counting with the header (Total C1 C2 C3) or the line that begins with “Number”. We do not see an error in the SF3B1 or TP53 data. Can the reviewer please specify what they thought was incorrect?
Check the Table 1 and the supplemental Table 1. Change commas by points in supplemental
RESPONSE: Tables. Done. We do not see any commas in what we submitted, so perhaps something got altered when the journal created their version from our original?
Discussion section: check line 241.
RESPONSE: We have changed this to read “patterns exist in CLL and that these are predictive" , adding the underlined words.
Revise the number of cells (line 322).
RESPONSE: The superscript has been changed to reflect the correct number of cells (1x104)
Reviewer 2 Report
The topic is very interesting and of great value to all hematologists. the manuscript is well written. However, few corrections could make it better.
1- after the introduction, you proceeded to the results. I needed to scroll down to search for the methodology to read first to help me understand. So, it is advisable to put the methodology and the statistical analysis before the results.
2- In table 1 you used median + - SD, it should be mean + - SD for normally distributed data.
3- In table 2, why did not you do multivariate for stage C or no major mutation though they are both significant in the univariate?
4- in the statistics part, Add the level of significance of p value.

Author Response
REVIEWER 2
The topic is very interesting and of great value to all hematologists. the manuscript is well written. However, few corrections could make it better.
1- after the introduction, you proceeded to the results. I needed to scroll down to search for the methodology to read first to help me understand. So, it is advisable to put the methodology and the statistical analysis before the results.
RESPONSE: We also prefer the traditional methods, intro results, discussion ordering, and wrote the manuscript as having the methods before the results. However the online style notes for the journal, puts it after the discussion (as in Nature journals) so that is how we submitted it. We will locate it wherever the editors tell us to.
2- In table 1 you used median + - SD, it should be mean + - SD for normally distributed data.
RESPONSE: Table 1 does display the mean and standard deviation as specified in the table legend. The label in the table needs to be corrected
3- In table 2, why did not you do multivariate for stage C or no major mutation though they are both significant in the univariate? .
RESPONSE: There was an error in the table and the values for Binet stage C were accidentally omitted. These have been added in. There was another error in that “No Major mutation” was listed,, but this was actually the baseline comparator and the numbers shown in the univariate section were actually those of “Major Mutation Present” . The variable name has been corrected and the multivariate data for “Major Mutation Present” was added in.
4- in the statistics part, Add the level of significance of p value.
RESPONSE: . We have noted using the commonly used p <= 0.05 threshold, but in other places, as noted we made adjustments for multiple searches.
Reviewer 3 Report
Interesting but need to experimental clarification
Author Response
REVIEWER#3
Interesting but need to experimental clarification
RESPONSE: There are no specific comments to respond to. We have asked the editors to ask this reviewer for clarification regarding what experimental clarification is desired so that we can address the issues. However we would note that the journal editors asked for a response within 5 days, which precludes conducting new experiments.